# Cooperative redox activation for carbon dioxide conversion

Zhong Lian[1,*], Dennis U. Nielsen[1,*], Anders T. Lindhardt[2], Kim Daasbjerg[1] & Troels Skrydstrup[1]

A longstanding challenge in production chemistry is the development of catalytic methods for the transformation of carbon dioxide into useful chemicals. Silane and borane promoted reductions can be fined-tuned to provide a number of C1-building blocks under mild conditions, but these approaches are limited because of the production of stoichiometric waste compounds. Here we report on the conversion of $CO_2$ with diaryldisilanes, which through cooperative redox activation generate carbon monoxide and a diaryldisiloxane that actively participate in a palladium-catalysed carbonylative Hiyama-Denmark coupling for the synthesis of an array of pharmaceutically relevant diarylketones. Thus the disilane reagent not only serves as the oxygen abstracting agent from $CO_2$, but the silicon-containing 'waste', produced through oxygen insertion into the Si–Si bond, participates as a reagent for the transmetalation step in the carbonylative coupling. Hence this concept of cooperative redox activation opens up for new avenues in the conversion of $CO_2$.

[1] Carbon Dioxide Activation Center (CADIAC), Interdisciplinary Nanoscience Center, Department of Chemistry, Aarhus University, Gustav Wieds Vej 14, 8000 Aarhus C, Denmark. [2] Carbon Dioxide Activation Center (CADIAC), Interdisciplinary Nanoscience Center, Department of Engineering, Aarhus University, Finlandsgade 22, 8200 Aarhus N, Denmark. * These authors contributed equally to this work. Correspondence and requests for materials should be addressed to T.S. (email: ts@chem.au.dk).

Carbon monoxide (CO) represents the most important one carbon building block on an industrial scene for the synthesis of value added bulk and fine chemicals including hydrocarbon fuels, aldehydes and carboxylates[1–4]. Its high toxicity and flammability has always been the sticky point in the handling and transportation of this gas, and therefore the identification of viable surrogates is welcomed. Indeed, several research groups have successfully disclosed the suitability of a number of different CO releasing reagents, such as aldehydes, alcohols and acid chlorides, but all have disadvantages with respect to atom efficiency and price[5–7]. Carbon dioxide represents another one carbon molecule being highly abundant but compared with CO it displays high kinetic and thermodynamic stability. Hence, the development of catalytic methods for the transformation of carbon dioxide into useful chemicals including carbon monoxide, formic acid, formaldehyde, methanol and others through the appropriate selection of the catalyst and reducing agent[8,9], is currently one of the most active research fields in chemical synthesis[10–13]. From an atom economy perspective, hydrogen represents the best reducing agent, but forcing conditions (high pressure, high temperatures) are generally required, and so far dihydrogen is produced mainly from non-sustainable carbon sources[14]. Instead silane and borane promoted reductions can be fine-tuned to provide these building blocks under mild conditions, but these approaches are significantly limited because of the production of stoichiometric waste compounds[15,16].

If suitable catalytic reaction conditions could be found for converting $CO_2$ to CO directly such chemistry would have significant impact on chemical synthesis. Seminal studies by the groups of Tominaga and Beller have demonstrated the potentiality of such an approach with *in situ* use of CO in hydroformylation and hydroxycarbonylation of olefins, but high reaction temperatures and in the former high hydrogen pressures are required[17,18]. Other studies have revealed that stoichiometric amounts of disilanes and diboranes are significantly more reactive towards the reduction of $CO_2$, even at room temperature, but such reactions are restricted by the waste generation[19–21].

In this paper, we describe our efforts on how to exploit the waste generated from the reduction of $CO_2$ with diaryldisilanes through its active participation in an ensuing reaction with the carbon monoxide formed. We demonstrate that diaryldisiloxane produced from the $CO_2$ reduction are excellent reagents for a Pd-catalysed carbonyative Hiyama–Denmark coupling generating diarylketones in good yields. Furthermore, the two transformations can be combined in a one-pot fashion and can be further exploited for the preparation of pharmaceutically relevant compounds as well as for selective carbon isotope-labelling. We believe this concept of cooperative redox activation whereby both reagents from the initial transformation undergo a second transformation will open up for new opportunities in $CO_2$ conversion.

**a** Reduction of $CO_2$ to carbon monoxide

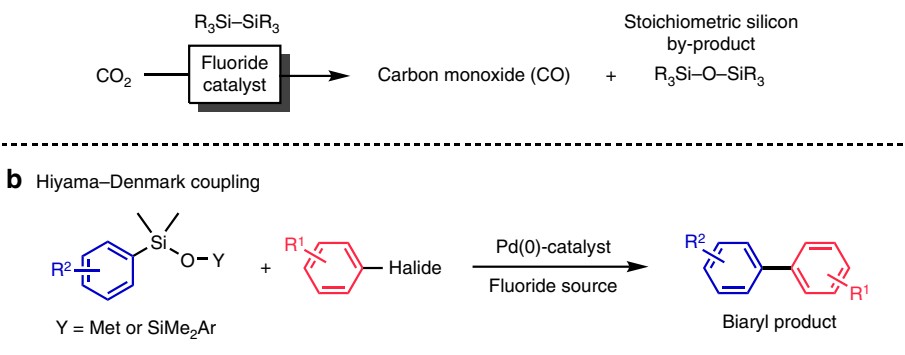

**b** Hiyama–Denmark coupling

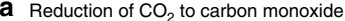

**c** Can cooperative redox activation of $CO_2$ be exploited for using waste disiloxane in a carbonylative coupling?

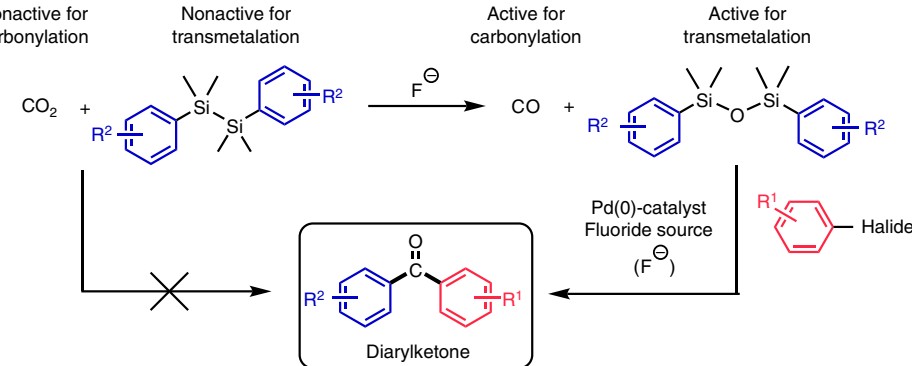

**Figure 1 | General concepts in cooperative redox activation with carbon dioxide.** Figure shows the overall strategy. (**a**) Reduction of $CO_2$ to carbon monoxide with disilanes produces stoichiometric silicon wastes. (**b**) Arylsilanolates and diaryldisiloxanes are viable reagents for their palladium-catalysed cross coupling with aryl halides to produce biaryls. (**c**) $CO_2$ and diarylsilanes cannot directly afford diarylketones in the presence of an aryl halide and a palladium catalyst. But through cooperative redox activation, these two reagents are transformed to carbon monoxide and a diarydisiloxane, which can participate in the three-component carbonylative coupling.

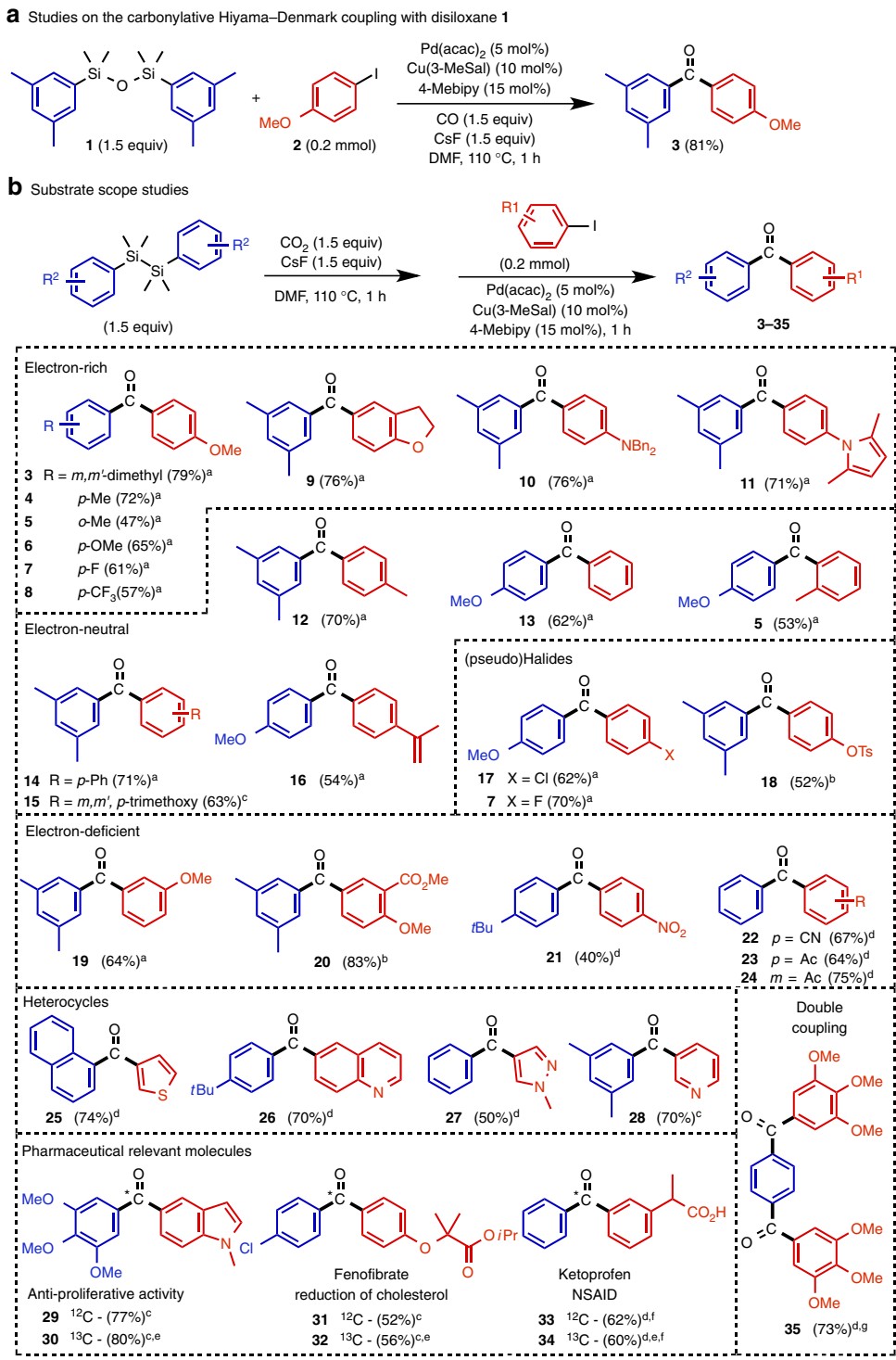

**Figure 2 | Substrate scope and applications to relevant pharmaceuticals possessing a diarylketone core.** The top row reaction shown in **a** reveals the reaction conditions identified for the carbonylative Hiyama-Denmark coupling between a diaryldisiloxane and an aryl halide. In **b** studies on the substrate scope are illustrated for the synthesis of diarylketones from an initial cooperative redox activation of $CO_2$ and a diaryldisilane to carbon monoxide and the corresponding diaryldisiloxane followed by the Pd-catalysed carbonylative cross coupling with an aryl iodide. All yields are based on an average of two runs.

## Results

**Preliminary considerations.** In 2014 we reported an expedient and room temperature reaction for the conversion of $CO_2$ to carbon monoxide with disilanes catalysed by fluoride salts

(Fig. 1a)[21]. We questioned whether the oxidized silicon waste product generated from the reduction of $CO_2$ to CO with a disilane could instead be viewed as a reagent for an ensuing second chemical transformation along with the produced CO.

**Figure 3 | Proposed mechanism for the transformation of CO$_2$ and diaryldisilanes to diarylketone.** Initially cesium fluoride (CsF) promotes the reduction of CO$_2$ to carbon monoxide and the oxidation of the diaryldisilane to the corresponding diaryldisiloxane (Ar, Aryl). Subsequent cleavage of the disiloxane by excess fluoride leads to the silanolate **38**, which enters *Cycle A* via ligand exchange with a Cu(I) complex. For the aryl halide, the catalytic *Cycle B* begins with oxidative addition of the Pd(0) species into the aryl halide bond generating a Pd(II)-aryl species that subsequently follows migratory insertion with carbon monoxide to form and acyl Pd(II) species **40**. Transmetalation with a Cu–silanolate and activation by a second silanolate promotes transmetalation to a diaryl Pd(II) complex, which reductively eliminates to the Pd(0) catalyst and the diarylketone.

We recognized that oxygen abstraction from CO$_2$ would generate a disiloxane, which if substituted with aryl groups would act as an aryl delivering agent in a Pd-catalysed Hiyama-Denmark coupling with a (hetero)aryl halide (Fig. 1b)[22,23]. Furthermore, we reasoned that if the CO could be exploited then non-symmetrical diarylketones would be made accessible for the first time through a modified three-component Hiyama-Denmark coupling in a single operation (Fig. 1c)[24]. Particularly noteworthy is the fact that neither CO$_2$ nor the disilane can participate in this latter transformation. Only through cooperative redox activation involving the reduction of CO$_2$ and oxidation of the disilane to a new set of reaction components can such a transformation take place. If successful, we recognized that this catalytic approach would open up for new applications in the utilization of CO$_2$ in chemical synthesis.

**Reaction discovery.** At the outset of our investigations, we examined the feasibility of a Pd-catalysed carbonylative Hiyama-Denmark coupling. Taking advantage of the extensive studies performed by the Denmark research group on the cross coupling of vinyl- and aryldisiloxanes or the corresponding silanolates with aryl/vinyl halides[25], we commenced work with the di-*m*-xylyldisiloxane and its carbonylative coupling with *p*-iodoanisole (**1** and **2** in Fig. 2a) in the presence of only a slight excess of CO generated in our two-chamber technology[26] (Supplementary Note 1). After an exhaustive optimization study (Supplementary Tables 1–4) we identified catalytic reaction conditions involving Pd(acac)$_2$ (acac, acetylacetonate) as the palladium source, 4,4'-dimethylbipyridine (4-Mebipy) as the ligand and finally cesium fluoride (CsF) and copper(I) 3-methyl salicylate (Cu(3-MeSal)) as promoters for the transmetalation step in DMF providing an excellent 81% yield of the diarylketone **3**. A reaction temperature of 110 °C was

required for full conversion, which is considerably higher than that observed for the normal Hiyama-Denmark coupling with aryl iodides, and which can be explained by an attenuated reactivity of the palladium catalyst with CO as a ligand[23].

We then proceeded to combine the carbonylative coupling with the CO$_2$ reduction step in the presence of a diaryldisilane. Hence 1.5 equiv. of CO$_2$ was injected into a single chamber reactor containing CsF and di-*m*-xylyldisilane, prepared from the arylation of dichlorotetra-methyldisilane with *m*-xylyl lithium[27]. After a short reaction time of 1 h, a DMF solution of the *p*-iodoanisole, containing Pd(acac)$_2$, 4-Mebipy and Cu(3-MeSal) was injected into the flask and heated to 110 °C for an additional hour generating the same diarylketone **3** in a satisfactory yield of 79% (Fig. 2b). We also observed the small formation of the direct coupling product (biaryl formation), as well as carboxylation of the aryl iodide. However, these two minor by-products are easily separated from the diarylketone in a single chromatographic step.

**Substrate scope.** Having demonstrated the feasibility of cooperative redox activation for diarylketone synthesis, we explored the scope of this transformation with respect to both the aryl iodide and diaryldisilane components. A variety of diaryldisilanes including substituents in the *p*-position of the arene unit (methyl, methoxy, fluoro, trifluoromethyl) were tested with *p*-iodoanisole, and pleasingly these provided adducts **4–8** (Fig. 2a, 57–72% yields), although a slight reduction in yield was noted with an *o*-positioned methyl group. Interestingly, the carbonylation yields were not influenced on going from electron rich (OMe, Me) to electron withdrawing substituents (F, CF$_3$). However, employing a disilane displaying a cyano group only provided the desired biarylketone in trace amounts (result not shown). Three other electron rich iodoarenes including

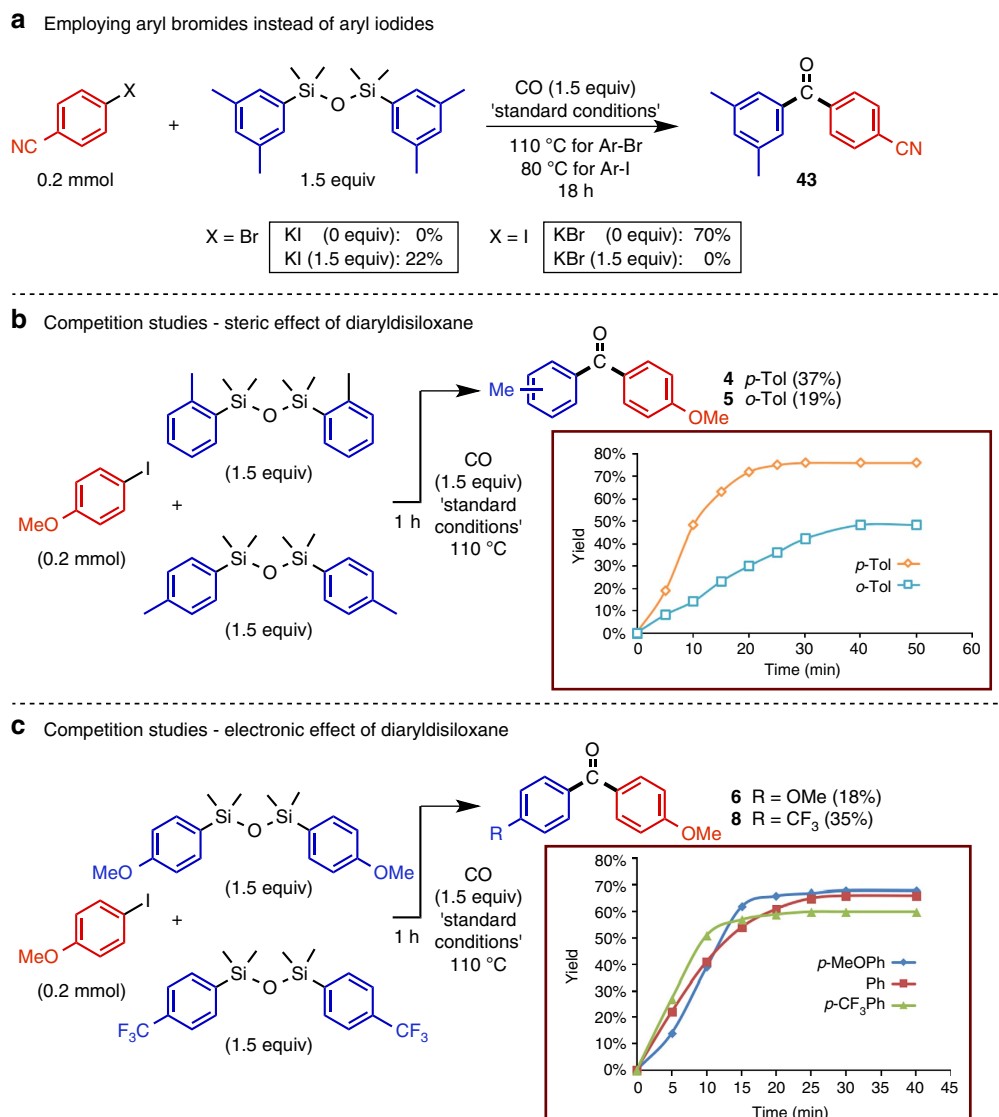

**Figure 4 | Substrate effects in the carbonylative Hiyama-Denmark coupling. (a)** Comparative coupling studies with an aryl bromide and iodide, and the effect of the addition of halide salts such as potassium bromide (KBr) and potassium iodide (KI) on the coupling efficiency. **(b)** The effect of substituent position on the diaryldisiloxane in both a competitive reaction whereby 1.5 equivalents of each disiloxane were reacted simultaneously with *p*-iodoanisole. The yields were determined after 1h of reaction. A similar ratio between the two coupling products was obtained after 30 min of reaction. In addition, the individual reactions were also tested (boxed graph) starting from the corresponding disilanes and $CO_2$. **(c)** A similar set of experiments as in B was carried out between an electron rich and electron poor diaryldisiloxane. The yields depicted in the graphs are based on GC using trimethoxybenzene as the internal standard.

*N,N*-dibenzyl-*p*-iodoaniline, 5-iodo-2,3-dihydrobenzofuran and 1-(4-iodophenyl)-2,5-dimethyl-1*H*-pyrrole coupled well with the di-*m*-xylyldisilane and $CO_2$ providing the corresponding diarylketones **9–11** in yields from 71 to 76% yield. Electron neutral aryl iodides proved worthy in their coupling with di-*m*-xylyldisilane and di-*p*-methoxyphenyldisilane (compounds **5, 12–16**, 53–71% yield). Aryl iodides displaying a second halide (Cl, F) or a pseudohalide such as a tosylate in the *p*-position were also compatible for these couplings providing a handle for further manipulations of the diarylketones (**7, 17** and **18**).

Considerable amounts of the biaryl adduct were formed with the electron deficient aryl electrophiles under standard conditions. However, by reducing the reaction temperature to under 110 °C provided acceptable coupling yields for substrates bearing functional groups such as an acetyl, nitro, cyano and carboxyl group (**20–24**, 40–83% yield), and thereby

demonstrating that an array of substituents are tolerated for this transformation. Heterocyclic iodides were amenable to the coupling conditions as well leading to ketones **25–28**, but here too lower reaction temperatures proved necessary for the good yields obtained. Interestingly, when the reaction temperature was lowered for the electron rich iodoarenes the corresponding benzoic acid derivatives were preferentially formed.

In a final demonstration, the concept of cooperative redox activation was adopted for the facile preparation of three pharmaceutically relevant molecules encompassing an anti-proliferative compound (**29**), (ref. 28) fenofibrate (**31**) (ref. 29) and ketoprofen (**33**) (ref. 30), all bearing a non-symmetrical diaryl ketone core in their structures. Particularly noteworthy were the possibilities to label these molecules (**30, 32** and **34**) with carbon-13 from the *in situ* preparation of $^{13}C$-$CO_2$ from $^{13}C$-labelled $BaCO_3$ (See Supporting Methods)[21].

Furthermore, using 1,4-diiodobenzene on a 0.1 mmol scale allowed for a double carbonylative coupling allowing the synthesis of biaryldiketone **35**.

**Mechanistic considerations and study**. A mechanistic proposal in line with the results produced is depicted in Fig. 3. Initial reduction of $CO_2$ by the diaryldisilane **36** generates CO and the diaryldisiloxane **37**, which enter their respective catalytic *Cycles A* and *B*. The disiloxane is cleaved in the presence of CsF providing an arylsilanolate **38** followed by a metal exchange process involving Cu(3-MeSal) in *Cycle A*. Subsequent transfer to a Pd–acyl complex **39** in *Cycle B*, generated from an initial oxidative addition of Pd(0) into the aryl halide bond followed by migratory insertion with the ligated CO, leads to the arylpalladium(II) arylsilanolate complex **40**. Whether this transfer occurs through the suggested copper-silanolate species or a related copper-aryl species is speculative. A second transmetalation step from Si to Pd finally generates the precursor **41** for the reductive elimination step to diarylketone formation. This step proceeds possibly via the intervention of an anionic species **40a** involving the activation of **40** with an additional silanolate in accord with the recent results from the Denmark group[31,32]. If transmetalation of **40** or **40a** in *Cycle B* is slow as is the case with electron rich aryl iodides run at lower reaction temperatures than 110 °C, then reductive elimination leads to the silylated carboxylate **42** (ref. 33). On the other hand, with electron poor aryl iodides biaryl bond formation becomes the major pathway but only at 110 °C. Lowering the reaction temperature allows aryl migratory insertion onto CO to compete with the transmetalation step (Si to Pd), and hence diarylketone formation dominates[34]. We assume here that the Si to Pd aryl transmetalation is faster in these cases than that for the electron rich aryl substrates because of the increased electrophilicity of the metal center in the acyl Pd complexes[35]. The proposed mechanistic cycle does not include the involvement of fluorosilicates, which cannot be ruled out.

Further mechanistic studies were undertaken to investigate the importance of the halide leaving group in the electrophile. As illustrated in Fig. 4a, attempted coupling of the (*p*-bromo) benzonitrile with di-*m*-xylyldisiloxane under standard reaction conditions led only to formation of *p*-cyanobenzoic acid. Partial conversion to the diarylketone **43** (22% yield) was nevertheless possible upon addition of 1.5–4 equiv. of KI, possibly achieved through a halide exchange process[36,37]. It is not clear why the aryl bromide leads exclusively to the carboxylic acid, but we speculate that after oxidative addition and migratory insertion, a fast reductive elimination provides an acyl bromide[38], which is subsequently attacked by the cesium silanolate. A similar effect was observed starting from (*p*-iodo)benzonitrile. Its coupling with di-*m*-xylyldisiloxane led to the isolation of the diarylketone **43** in a 70% yield; however in the presence of KBr only the formation of *p*-cyanobenzoic acid was obtained.

Two competition studies were performed to delineate both electronic and steric effects of the diaryldisiloxane on the carbonylative coupling with *p*-iodoanisole. In the first experiment, the diarylketones **4** and **5** were compared from the competitive coupling of di-*p*-tolyldisiloxane and di-*o*-tolyldisiloxane after 1 h (Fig. 4b), revealing the former to be more reactive. Similar results were obtained also after 30 min. Use of larger equivalents (5–10) of the disiloxanes led to significant decomposition. These results were also confirmed from comparison of the individual reactions starting initially from the two disilanes with $CO_2$ (Graph in Fig. 4b). The sterically more demanding di-*o*-tolyldisiloxane generated from the disilane was again inferior to that of the *para*-counterpart with substantial amounts of the *p*-methoxybenzoic acid formed, implying that transmetalation from Si to Pd is slow thereby favoring direct reductive elimination to the silylated carboxylate.

In the final set of experiments we examined the reactivity between two diaryldisiloxanes bearing either an electron donating ($OCH_3$) or an electron withdrawing substituent ($CF_3$) in the *para*-position. Performing the competitive carbonylative coupling experiment with *p*-iodoanisole as above (Fig. 4c), the yield for the diarylketone **8** ($CF_3$) was approx. twice that for the ketone **6** ($OCH_3$). Similarly for the individual reactions after 5 min, the yield from the more electron deficient disiloxane was approx. twice that for the electron rich derivative (Graph in Fig. 4c). Denmark and co-workers have shown that transmetalation is more efficient for electron-rich aryl groups on the silicon atom, which seems to contrast these results[36]. Therefore, our observations should be interpreted as a result of the faster formation of the (*p*-trifluoromethyl)phenylsilanolate in comparison to (*p*-methoxy)phenylsilanolate because of greater electrophilicity at the silicon center on the di(*p*-trifluoromethyl)phenyldisiloxane and thereby its greater propensity to react with CsF.

Altogether the results represent an important demonstration on how cooperative redox activation can provide an alternative route for the conversion of carbon dioxide into useful chemicals. We anticipate that this concept will find other uses in synthetic organic chemistry and will contribute to the development of chemical transformations whereby waste byproducts containing valuable groups can find useful applications.

## Methods

A flame-dried COtube (Supplementary Methods) charged with a stirring bar and disilane (0.3 mmol, 1.5 equiv) was transferred to an argon-filled glovebox. CsF (45.6 mg, 0.3 mmol, 1.5 equiv) and DMF (1.0 ml) was then added in that order. The COtube was sealed with a screwcap fitted with a Teflon seal and removed from the glovebox. $CO_2$ (7.2 ml, 0.3 mmol, 1.5 equiv) was then injected via syringe and the reaction mixture was stirred at the stated temperature for 1 h. Meanwhile, a stock solution (3 equiv) was made which was used to provide dublicates of the reaction in the following manner: Aryl iodide (0.6 mmol, 3.0 equiv) and 4,4-dimethyl-2,2-bipyridine (16.5 mg, 0.090 mmol) were added to a flame-dried 8 ml vial charged with a stirring bar and transferred to an argon-filled glovebox together with two flame-dried 4 ml vials. Pd(acac)$_2$ (9.0 mg, 0.030 mmol) and copper(I) 3-methylsalicylate (12.9 mg, 0.060 mmol) were transferred to seperate 4 ml vials and dissolved in DMF (750 μl), respectively. The vials were briefly shaken to obtain a homogenous solution. DMF (1.5 ml) was added to the 8 ml vial followed by heating at 90 °C for 10 s. The vial was then removed from the heating block and the Pd- and Cu solutions were then added. The 8 ml vial was then capped with Teflon-containing screw-cap and removed from the glovebox. 1 ml of the stock solution was then injected to the COtube via syringe and reacted for 1 h. The reaction was cooled to room temperature, diluted and transferred with EtOAc to a 25 ml flask and the volatiles were removed *in vacuo*. The crude residue was subjected to flash column chromatography using pentane/ethyl acetate as eluent to afford the desired product. For NMR spectra and specific conditions for compounds **3–35** and **43**, Supplementary Figs 1–36 and Supplementary Methods.

**Data availability**. The data supporting this work are available from the authors on reasonable request.

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

## Acknowledgements

We thank the Danish National Research Foundation (grant no. DNRF118), the Villum Foundation, the Danish Council for Independent Research: Technology and Production Science and Aarhus University for financial support. Z.L. thanks the Chinese Scholarship Council for a Graduate Fellowship. We thank S. Kramer for discussions.

## Author contributions

Z.L. and D.U.N. contributed equally to this study. Z.L. and D.U.N designed and performed the optimization, scope and mechanistic experiments. All authors analysed and interpreted data from these experiments. T.S. conceived and coordinated the study, and also wrote the paper with input or editing from all authors.

## Additional information

**Competing financial interests:** A.T.L. and T.S are co-owners of SyTracks a/s, which commercialises COtube, COware and COgen. The remaining authors declare no competing financial interests.

