## [Peer review file · Nature Communications]

Reviewers' comments:

Reviewer #1 (Remarks to the Author):

This manuscript from Skrykstrup et al. describes an elegantly engineered conversion of carbon dioxide and diaryldisilanes into biaryl ketones and polymethylsiloxane byproducts. The authors have combined their previous observation that disilanes can deoxygenate carbon dioxide to form disiloxanes and carbon monoxide in the presence of a fluoride activator. In this work, the authors proposed to combine those two byproducts of CO₂ reduction following the prior work from Hiyama and Denmark on the palladium catalyzed cross-coupling of silicon derivatives. After an extensive optimization (described in the Supporting Information) the authors identify reaction conditions that allow the through process, i.e. starting with CO₂, diaryldisilanes, aryl iodides and a combination of both palladium and copper catalysts in the presence of cesium fluoride to generate unsymmetrical biaryl ketones. Figure 2 contains a collection of the successful coupling products derived from a modest selection of aryl iodides and an even more limited set of diaryldisilanes. The overall yields of diaryl ketones are modest to good considering the number of steps involved in the transformation. The authors also illustrate the ability to incorporate C-13 label into the ketone through the use of labelled CO₂ generated from labelled barium carbonate.

Given the number of steps involved in the simultaneously operating catalytic cycles, it is not surprising that some optimization was needed for individual cases to suppress undesired biaryl coupling and reductive elimination to aryl carboxylates. The final aspect of the preparative section presents the syntheses of a number of therapeutic agents containing a biaryl ketone motif.

The next section describes the author's efforts to provide a mechanistic understanding of the overall transformation. They propose the generation of the two reactants, carbon monoxide and a the disiloxanes in an initial, irreversible step, followed by the engagement of two simultaneously operating catalytic cycles. The first cycle (A) involves the metal exchange of the initially generated cesium silanolate to a copper silanolate which enters the second catalytic cycle (B) whereupon it undergoes a second metal exchange to form a palladium silanolate from the acylpalladium species created earlier in the second cycle. It is not apparent from the text or from the optimization why the authors propose this particular role for the copper co-catalyst. Certainly other options are possible including the direct transmetalation of the copper silanolate to form an aryl copper species that could then transmetalate again to form the arylpalladium acyl complex. The authors adapt their mechanistic proposal to the mechanistic foundations of biaryl cross-coupling established recently by Denmark et al. Although these assumptions are very reasonable, the use of cesium fluoride in this reaction manifold may open different reaction pathways involving fluorosilicates.

The authors also identify a very strict requirement for the use of aryl iodides, not aryl bromides likely because of a rapid reductive elimination of the arylpalladium acyl bromide to form an acid bromide that reacts irreversibly with silanolate. Although the use of aryl bromides in the presence of 1.5 equiv of KI did lead to some productive reaction, it does not make the process of practical utility. This is an unfortunate limitation. Additional competition studies revealed a significant rate effect on

the substituents on the diaryl disiloxanes. 2-Tolyl reagents reacted more slowly than 4-tolyl reagents and 4-methoxyaryl disiloxanes reacted more slowly than 4-trifluoromethylaryl disiloxanes which the authors interpret as slower generation of the requisite reagent. This trend is opposite of that found by Denmark in reactions operating under turnover-limiting transmetalation. Given the use of only 1.5 equiv of cesium fluoride and the unproductive consumption of some of the disiloxanes as a silyl fluoride, these reactions are not likely operating at saturation of the palladium catalyst. Thus it is also possible that the reason for faster reaction of the 4-trifluoromethylaryl disiloxanes is the higher concentration of the activated form arising from the higher electrophilicity of the silicon atom in this species. DeShong has also reached this erroneous conclusion in a Hammett study which did not reflect the intrinsic reactivity of the activated complexes (i.e., the 4-methoxyaryl species is actually more reactive).

Overall, this contribution represents a very clever concept for the efficient use of CO₂. Although this process in no way will ameliorate the problem of atmospheric CO₂ or the energy required for hydrogen generation by using disilanes as reducing agents, it is a creative piece of reaction engineering and has proven the ability to incorporate CO₂ in a new and productive fashion. Preparatively however, the process does suffer from a number of significant limitations, i.e. the requirement for the use of aryl iodides, the requirement to prepare the disilanes, the need to optimize conditions to suppress competing processes. The authors have provided a framework for understanding the mechanistic features of the reaction, but certainly more is needed.

In the opinion of this reviewer, the authors have cleverly chosen the venue for this publication as Nature and its many offspring value novelty and splash over practicality. Thus, with appropriate attention to the editorial comments on the attached, edited pdf version, the manuscript is suitable for publication in this journal.

Reviewer #2 (Remarks to the Author):

This work by the Skrydstrup group describes an intriguing Pd-catalyzed cross-coupling carbonylation technique using carbon dioxide as carbon monoxide surrogate by utilizing bis-silanes as reducing agents. While the means to reduce CO₂ to CO by bis-silanes is a known process, such a technique inevitably ends up in stoichiometric silanol derivatives that cannot be employed for further transformations, thus not representing an added value from an atom-economical standpoint. This work demonstrates that a Hiyama-Denmark carbonylation protocol can be within reach by using the silanols that derive from the reduction of CO₂ to CO, thus constituting an integrated catalytic technique that follows the principle of both atom- and step-economy while not requiring the utilization of toxic CO for promoting carbonylation events. The authors came up with a catalytic cycle that virtually explains all results obtained, including the corresponding side-reactions. Additionally, competitive experiments were performed to delineate the steric and electronic effects of the in situ generated diaryl disiloxane.

As nicely illustrated in Figure 2, a wide variety of aryl iodides, regardless of their electronic or steric properties could be coupled with equal ease, obtaining in virtually all cases analyzed good yields of

the final products. Although one might argue that this reaction is inherently limited to aryl iodides, this transformation is certainly unique and opens up new vistas in the utilization of CO₂ as CO surrogate while integrating a catalytic cycle for promoting two C-C bond-formations with an exquisite atom- and step-economy. That being set, I am absolutely confident that this publication will attract immediate interest at the Community, offering a conceptually new technique that will likely be rapidly embraced by practitioners in the field. Therefore, I would strongly recommend acceptance of this manuscript in Nature Communications pending minor modifications.

- (1) What is the mechanistic rationale behind the observed biaryl product with electron-withdrawing aryl electrophiles ?
- (2) The scope of these reactions is apparently limited to a rather specific class of diaryldisilanes. It would be interesting to include additional examples with diaryldisilanes bearing functional groups to explore the potential of this technique
- (3) I would recommend the authors to include a Hammett Plott with differently substituted aryl bromides.
- (4) Apparently, electron-withdrawing aryl bromides can't be utilized due to the formation of side-products, mainly biaryl formation and benzoic acid. What about electron-neutral or electron-rich aryl bromides ? Is there any rationale for the lack of reactivity associated to the utilization of phosphine-type ligands ?
- (5) Although not necessarily required, the inclusion of vinyl electrophiles would constitute an added value.
- (6) Is it possible to promote a double carboxylation with, for example, 1,3-diiodobenzene ?

Reviewer #3 (Remarks to the Author):

The manuscript "Cooperative redox activation for carbon dioxide conversion" by Skrydstrup and co-workers describes a one-pot, two-step process that combines the reduction of carbon dioxide (CO₂) with diaryldisilanes to generate carbon monoxide (CO) with a Pd/Cu-catalyzed carbonylative arylation process to obtain unsymmetrical diaryl ketones. The key concept touted by the authors is the so-called "cooperative redox activation," in which CO₂ and diaryldisilane reacts to generate the necessary products for the carbonylative arylation reaction. In this paper, the authors found the conditions to enable the carbonylative arylation process (Fig. 2A) in a two-chambered system, and then applied the same conditions for the one-pot, two-step process (Fig. 2B). Their substrate scope is relatively good, and a wide range of diaryl ketones were obtained in good yields.

While the work described by the authors is interesting, it is the opinion of this reviewer that the manuscript is not appropriate for a general scientific journal such as Nature Communications. The major reason for this decision is rooted in the belief that this work does not possess sufficient novelty or insight into chemistry to be of interest to readers not in the field of organic chemistry. Basically, the individual steps of this two-step process have already been reported, and the authors combined them in their studies. While the idea of using a waste co-product in a subsequent step

may not have been applied in organic transformations with CO₂, such a concept has been utilized in other sequential reactions (Angew. Chem. Int. Ed. 2010, 49, 4976).

Response to the reviewer comments

All changes are highlighted in yellow in the manuscript.

(Q/C: Question/Comment. A: Answer)

Reviewer 1:

Comments/questions from the attached PDF.

Q/C: How are the disilanes generated?

A: The disilanes are formed from lithiation of an aryl bromide using BuLi followed by addition of 1,2-dichlorotetramethyldisilane. This is described under Figure 2 and in the Supporting Information section.

Q/C: Remove "scene" and replace with "scale" (Page 1, first sentence after the abstract).

A: Correction made.

Q/C: Please add review by Beller and paper from Hull.

A: Both references have been added.

Q/C: What is the exact stoichiometry of the reaction? (Figure 1C)

A: The stoichiometry is already depicted in Figure 2.

Q/C: What is the role of the 4-Mebipy ligand?

A: We suspect the ligand to be coordinating to Pd and Cu throughout the reaction to increase the stability and reactivity of the catalysts.

Q/C: "These two are NOT electron deficient"

A: The methoxy group has an electron-withdrawing effect when placed in the *meta*-position and an electron-donating effect when positioned at the *para*-position. This is why we classified compounds **18** and **19** as "electron-deficient". However, the *para*-methoxy group could be argued to cancel out the effect of the two *meta*-positioned

methoxy groups. The trimethoxy-substituted biarylketone has therefore been moved to the "electron-neutral" section.

Q/C: Comments on yield variation?

A: Electron-rich and electron-neutral iodides worked best at 110 degrees whereas more electron-deficient aryl iodides required lower temperature due to competing biaryl formation. The yields are in general around 60–70%, whereas the remaining iodide is converted into the corresponding biaryl or carboxylic acid.

Q/C: General comments regarding Figure 3.

A: All suggested corrections to the figure have been implemented. We currently do not have any evidence for the role of the copper. The suggested role is purely speculative. We cannot rule out the formation of an aryl-copper species instead of the copper-silanolate species. This had been added to the manuscript.

Q/C: Use another term for the formation of the copper-silanolate.

A: "Transmetallation" has been replaced by "ligand exchange". The remaining suggestions regarding the text for Figure 3 have been made.

Q/C: Are the yields depicted in the graphs in Figure 4 based on isolation or internal standard?

A: All yields in these graphs are calculated from GC using trimethoxybenzene as an internal standard. This has been added to the manuscript.

Q/C: "Not sure that this is a relevant comparison after 5 minutes... looks like a short induction period. They read the same point at nearly the same time".

A: The effect of the electronics are smaller compared to that of the steric nature of the disilane (Figure 4B vs Figure 4C). The reaction is essentially done after 20 minutes and therefore the induction period will be short.

Additional questions/comments from Reviewer 1:

Q/C: *"... the use of cesium fluoride in this reaction manifold may open different reaction pathways involving fluorosilicates."*

A: This has been added to the manuscript.

Q/C: *2-Tolyl reagents reacted more slowly than 4-tolyl reagents and 4-methoxyaryl disiloxanes reacted more slowly than 4-trifluoromethylaryl disiloxanes which the authors interpret as slower generation of the requisite reagent. This trend is opposite of that found by Denmark in reactions operating under turnover-limiting transmetalation. Given the use of only 1.5 equiv of cesium fluoride and the unproductive consumption of some of the disiloxanes as a silyl fluoride, these reactions are not likely operating at saturation of the palladium catalyst. Thus it is also possible that the reason for faster reaction of the 4-trifluoromethylaryl disiloxanes is the higher concentration of the activated form arising from the higher electrophilicity of the silicon atom in this species. DeShong has also reached this erroneous conclusion in a Hammett study which did not reflect the intrinsic reactivity of the activated complexes (i.e., the 4-methoxyaryl species is actually more reactive).*

A: We do not state that the electron-poor aryl groups on the silicon migrates faster than the electron-rich aryl groups. In fact, we reference the paper in which the Denmark group demonstrates that the opposite holds true (reference 35). We propose that the enhanced reactivity is due to the higher concentration of the electron-poor aryl silanolate due to the increased electrophilicity of the silicon, which the reviewer also points out. This has been further underlined in the manuscript.

Reviewer 2:

Q/C: *What is the mechanistic rationale behind the observed biaryl product with electron-withdrawing aryl electrophiles ?*

A: Aryl migration to the Pd-coordinated CO is slow for electron-poor aryl groups. Therefore, CO-insertion is omitted for these electrophiles and biaryl compounds becomes the main product. This is also stated in the manuscript (See reference 35).

Q/C: *The scope of these reactions is apparently limited to a rather specific class of diaryldisilanes. It would be interesting to include additional examples with diaryldisilanes bearing functional groups to explore the potential of this technique.*

A: This would be a nice addition to the reaction. The disilanes are prepared via lithiation of aryl bromides, which limits the functional group tolerance displayed by the aryl group. Nevertheless, a disilane carrying *p*-CN substituted aryl groups was prepared and

tested. Unfortunately, the desired biarylketone was only synthesized in a low yield. Hence, the use of these disilanes requires further optimization. A sentence has been added in the manuscript describing this experiment.

Q/C: I would recommend the authors to include a Hammett Plott with differently substituted aryl bromides.

A: We assume that the referee means aryl iodides. Since the reactivity of the aryl iodides are very temperature dependent (electron-rich requires a reaction temperature of 110 degrees, whereas electron-poor aryl iodides give the most optimal yield of the diaryl ketones at lower temperatures, 80–100 degrees), the information obtained from creating a Hammett plot would therefore be misleading. For this reason, we have not performed a Hammett analysis

Q/C: Apparently, electron-withdrawing aryl bromides can't be utilized due to the formation of side-products, mainly biaryl formation and benzoic acid. What about electron-neutral or electron-rich aryl bromides?

A: We attempted to use bromobenzene and 4-bromoanisole, however, no reaction was observed. Only starting material could be detected by analysis via ^1H NMR spectroscopy, suggesting that the oxidative addition into these compounds is ineffective. Definitely further work will be carried out to attempt to identify suitable conditions, but this will require considerably more experimentation.

Q/C: Is there any rationale for the lack of reactivity associated to the utilization of phosphine-type ligands?

A: Phosphine ligands are more common ligands for palladium catalysis rather than bipyridine ligands, so it is a good question. Phosphines are more electron-rich compared to bipyridines, which would increase the electron density on the Pd-center. This might interfere with the Si to Pd transfer of the aryl group, as electron-rich aryl groups generally migrate faster. If this step is slow, then the reductive elimination to the acylated silanolate might be favoured. Furthermore, the larger bite angle displayed by the phosphine ligands would also increase the rate for the reductive elimination. The corresponding carboxylic acid was the major product formed when evaluating phosphine ligands, which would support this theory.

Q/C: Although not necessarily required, the inclusion of vinyl electrophiles would constitute an added value.

A: Different vinyl electrophiles were attempted as coupling partners, however, no product was observed. Furthermore, stilbene-substituted disilanes/disiloxanes were evaluated, but again, no product was formed. We currently do not know why, this is the case.

Q/C: Is it possible to promote a double carboxylation with, for example, 1,3-diiodobenzene?

A: Yes, indeed it was. Using 1,4-diiodobenzene on a 0.1 mmol scale with a trimethoxy-substituted diaryldisilane allowed for the isolation of the double coupled product in 73% yield. This has been added to the manuscript.

Reviewer 3:

Q/C: While the work described by the authors is interesting, it is the opinion of this reviewer that the manuscript is not appropriate for a general scientific journal such as Nature Communications. The major reason for this decision is rooted in the belief that this work does not possess sufficient novelty or insight into chemistry to be of interest to readers not in the field of organic chemistry. Basically, the individual steps of this two-step process have already been reported, and the authors combined them in their studies. While the idea of using a waste co-product in a subsequent step may not have been applied in organic transformations with CO₂, such a concept has been utilized in other sequential reactions (Angew. Chem. Int. Ed. 2010, 49, 4976).

A: Reviewer 3 argues that the manuscript lacks novelty and that the two-step process has already been reported. Although the first step (reduction of CO₂ to CO using a disilane to form a disiloxane) has been reported by us earlier, the ensuing carbonylative Hiyama-Denmark reaction *is not known* in the literature. Therefore, considerable efforts were required to develop this reaction. In fact it is amazing that this carbonylative reaction even works as the major concern would be siloxycarbonylation of the aryl halide to form O-silyl benzoates. Next, we were able after careful design to combine the two transformations, so they could go hand in hand and thereby convert CO₂ to something pharmaceutically useful.

Reviewer 3 also makes reference to a paper whereby the Wittig reaction is performed and the waste phosphine oxide is exploited as a mediator for an ensuing silane reduction of the carbon-carbon double bond formed. Where I can see the idea of the waste being used, in this particular example the waste is exploited as a catalyst for the next reaction. In our case, we exploit the waste to become part of the product being the diarylketone. So these two approaches are conceptionally different. We believe that our method is unique and of higher value for waste usage particularly for the conversion of CO₂.